# Advanced Anterior Eye Segment Imaging for Ichthyosis

**DOI:** 10.3390/jcm12186006

**Published:** 2023-09-16

**Authors:** Anna Micińska, Anna Nowińska, Sławomir Teper, Joanna Kokot-Lesik, Edward Wylęgała

**Affiliations:** 1Ophthalmology Department, District Railway Hospital, 40-760 Katowice, Poland; 2Chair and Clinical Department of Ophthalmology, Faculty of Medical Sciences in Zabrze, Medical University of Silesia, 40-055 Katowice, Poland

**Keywords:** ichthyosis, cornea, dystrophy, ocular surface, optical coherence tomography, confocal microscopy

## Abstract

The purpose of this study was to describe ocular surface and anterior eye segment findings in various types of ichthyoses. Methods: This was a single-center prospective observational study. The study group consisted of five patients (P1–P5) aged 13–66 years. Multimodal imaging was performed, including slit-lamp examinations, swept-source optical coherence tomography (SS-OCT), and in vivo confocal microscopy (IVCM). Results: All patients were diagnosed with moderate-to-severe dry eye disease (DED). The corneas showed a significant pattern of irregularity, with a significant difference between the corneal thickness at the apex (CAT) and the corneal thinnest thickness (CTT), exceeding 375 µm. Three patients were diagnosed with ectasia patterns based on SS-OCT. All patients showed abnormalities in at least one Fourier index parameter for at least one eye at 3 or 6 mm in the keratometric, anterior, or posterior analyses. IVCM examinations revealed changes in all corneal layers. Conclusions: By combining the results of multimodal imaging, we were able to detect preclinical abnormalities, distinguish characteristic changes common to ichthyosis, and reveal the depth and characteristics of corneal abnormalities. Therefore, patients with ichthyosis should be examined for DED and ectatic disorders early in clinical practice.

## 1. Introduction

Ichthyoses are a group of heterogeneous inborn diseases with the primary feature of a defective epidermal barrier that causes hyperkeratosis, skin scaling, and inflammation, either affecting the skin only (non-syndromic) or being associated with internal organ disorders (syndromic). Further categorization is based on the manifestations at birth (congenital forms) or during the first year of life (vulgar forms). In 2009, the First Ichthyosis Consensus Conference established an international consensus on the nomenclature and classification of inherited ichthyoses [1]. Further advances in diagnosis and treatment are primarily related to advances in clinical diagnostic tools, molecular testing, and genetic therapy [2,3,4]. Ocular involvement may be regarded as a significant feature of ichthyoses; however, the manifestation and severity of ocular signs depend on and are related to the clinical form of ichthyosis. The prevalence of specific forms and, as a result, ocular surface involvement vary significantly among ichthyoses. According to prevalence, the general characteristics of ichthyoses are presented.

Common forms of ichthyosis, ichthyosis vulgaris (IV), and recessive X-linked ichthyosis (RXLI), are characterized by a prevalence of 1:250–1000 and 1:2000–6000, respectively [1,4,5,6,7]. Form IV (OMIM 146700), which shows semi-dominant inheritance, is the mildest form, characterized by scaling, xerosis, pruritus, and eczema. Symptoms usually manifest in the first few years of life. Form IV is characterized by frequent atopic manifestations [1,2,4]. RXLI (OMIM 308100) is the second most frequent type of ichthyosis and represents a benign form of ichthyosis that affects men almost exclusively. It develops because of the accumulation of undegraded cholesterol sulfate, which is responsible for scale formation during steroid sulfatase deficiency [1,2,4]. RXLI is associated with the occurrence of pre-Descemet’s membrane corneal dystrophy (PDCD), which is described in 50% of patients [1]. Dystrophy is characterized by the accumulation of polymorphic grey hyperreflective opacities located anterior to Descemet’s membrane and revealed by slit-lamp examination. In vivo confocal microscopy (IVCM) reveals enlarged hyperreflective keratocytes with extracellular deposits in the posterior corneal stroma [8,9,10,11]. 

Autosomal recessive congenital ichthyosis (ARCI) is a heterogeneous group of recurrent inherited disorders with congenital ichthyosis but no extracutaneous involvement [1,3]. Its prevalence is estimated to be 1 in 100,000 individuals. ARCI refers to harlequin ichthyosis (HI), lamellar ichthyosis (LI), and congenital ichthyosiform erythroderma (CIE). Most phenotypes are severe; however, minor variants exist, such as the self-healing collodion baby (SHCB) and bathing suit ichthyosis (BSI). ARCI is caused by mutations in more than a dozen different genes, including TGM1 (Transglutaminase-1), ABCA12 (ATP-binding cassette sub-family A member 12), CYP4F22 (Cytochrome P450 4F22), ALOXE3/ALOX12B (epidermal lipoxygenase-3/12R-lipoxygenase), NIPAL4 (magnesium transporter NIPA4), CERS3 (ceramide synthase-3), SDR9C7 (short-chain dehydrogenase/reductase family 9C member 7), PNPLA1 (patatin-like phospholipase domain-containing protein 1), SLC27A4 (long-chain fatty acid transport protein 4), and LIPN (epidermal lipase N) [4]. 

HI (OMIM 242500) is a severe condition that is associated with increased perinatal mortality. The clinical features of HI include thick gray or yellowish scales with severe collodion membranes, extreme ectropion and eclabium, contractures, a broadened nose, and synechiae of the auricles and, sometimes, toes. HI is caused by a loss-of-function mutation in the adenosine triphosphate (ATP)-binding cassette subfamily A member 12 (ABCA12), which leads to lipid transport disruption in keratinizing keratinocytes in the upper epidermis [1,4,12]. The LI phenotype (OMIM 242300) can be milder than the harlequin phenotype and phenotypic heterogeneity may exist. Generally, the phenotype is characterized by generalized large brownish or dark scales, often combined with palmoplantar keratoderma, ectropion, and anhidrosis [1,2,3,4]. Bathing suit ichthyosis (BSI) is a minor variant of LI that is caused by mutations in TGM1. Patients develop a scaling pattern that only affects the trunk and warmer skin areas, such as the axillary region or scalp [13,14].

Although each variant of ichthyosis has its own characteristics and differing intensities of abnormalities, ocular involvement is regarded as a major feature of the disease. A broad range of ocular findings is described in the literature, mainly involving the ocular surface and anterior eye segment, but there are limited reports on the association of ichthyosis with glaucoma, optic neuropathy, coloboma of the iris, choroid, and retina, and crystalline macular dystrophy [15,16,17].

Ocular surface involvement includes a wide variety of symptoms, such as the scaling of the eyelids, cicatricial ectropion, madarosis, entropion, chronic keratoconjunctivitis, Meibomian gland dysfunction, epithelial corneal defects, punctate keratitis, corneal ulcers, scarring and perforation, limbal stem cell deficiency and neovascularization, band keratopathy, PDCD, corneal irregularity, thinning, and keratoconus [16,17,18,19,20]. 

Ocular findings in ichthyosis are usually based on slit-lamp examinations, relatively small patient sample sizes, or case reports. The knowledge regarding ocular complications is also limited due to the extreme rarity of particular ichthyosis variants, as well as the limited lifespans of patients affected with syndromic forms. The slit-lamp examination does not provide an in-depth examination of the ocular surface and the corneal layer; therefore, the current knowledge on ocular complications may be limited. 

In recent years, multimodal imaging in ophthalmology, combining topography, pachymetry, corneal biomechanics, and morphology data, has provided significant new insights into the diagnosis of ocular surface, corneal, and anterior eye segment diseases, especially in cases of rare congenital diseases, such as corneal dystrophies [8,21,22]. The experience gained based on corneal dystrophy analysis may be utilized in ichthyoses to provide comprehensive assessments in terms of topography, morphology, and cellular imaging. To date, multimodal imaging has only been described for specific ichthyosis-related syndromes, such as PPCD [8,9,10,11]. Describing the morphological features of ichthyoses, including the character of abnormalities, anterior and posterior keratometry values, and the morphology of all corneal layers, using non-invasive imaging methods, would produce new valuable data in the context of the impact of ichthyoses on the corneal structure.

Two imaging techniques play a crucial role in ocular surface assessments: optical coherence tomography and confocal microscopy. Anterior eye segment optical coherence tomography (AS-OCT) is a precise, contactless technique for obtaining high-resolution ocular tissue images. Swept-source optical coherence tomography (SS-OCT), introduced in 2005, is a next-generation Fourier-domain OCT with a scanning speed of 50,000 A-scans/s and an axial resolution of 10 μm. It allows for the observation of anterior eye segment morphology, as well as analyses of topography, pachymetry, and Fourier indices (FI) [23,24,25]. To date, no published data are available regarding SS-OCT findings in patients with ichthyosis.

In vivo confocal microscopy (IVCM) enables in vivo corneal imaging with an axial resolution of 1 μm. IVCM is widely used to analyze the microscopic structure of the corneal layers, from the epithelium to the endothelium. This technique is useful for several clinical conditions, including infectious keratitis, dry eye disease (DED), corneal dystrophies, and degenerations [26,27]. To date, the use of IVCM in ichthyoses was limited to PPCD in RXLI [8,9,10,11].

We present a report on the clinical features of patients with various ichthyoses based on multimodal imaging, including findings from ocular surface assessments, SS-OCT, and IVCM. By combining these imaging methods, we aimed to characterize the appearance of the ocular surface, determine the depth of changes, and provide a deeper understanding of the features associated with different variants of ichthyosis.

## 2. Materials and Methods

### 2.1. Study Design

This single-center, prospective, observational study was conducted in accordance with the principles of the Declaration of Helsinki. The Bioethical Commission of Silesian Medical University in Katowice, Poland (KNE/0022/KB1/43/I/14), approved the study protocol. After providing written informed consent, the participants qualified for the research project. Written informed consent was obtained from the parents of participants aged <18 years. Patients were able to withdraw from the study at any time. All study participants were covered by the hospital’s third-party liability insurance. All examinations were conducted in accordance with the principles of good clinical practice applicable at the Chair and Clinical Department of Ophthalmology, Faculty of Medical Sciences in Zabrze, Medical University of Silesia, Katowice, Poland. The appropriate anonymization of personal information in the files was based on the assigned patient ID number. 

The primary purpose of this study was to describe ocular surface and anterior eye segment findings and analyze the results of multimodal imaging, including slit-lamp examinations, SS-OCT, and IVCM, in patients with different clinical types of ichthyosis.

### 2.2. Study Participants

The study group consisted of patients referred from a dermatological clinic or a general practitioner. The inclusion criteria were a diagnosis of ichthyosis, which was assessed based on the results of the dermatological consultation provided by patients and preferably supported by genetic testing. The exclusion criteria were: ophthalmic or systemic diseases other than ichthyosis, which are known to affect the ocular surface; previous eye injury; eye surgery; and the use of medication that is proven to affect the cornea. Examples of such ocular diseases include corneal dystrophies (except PPCD) and degenerations, uveitis, allergic conjunctivitis, active infectious keratitis, scleritis, and ocular tumors. Examples of surgeries include keratoplasty, cataract surgery, and pars plana vitrectomy. Examples of systemic diseases include Graves orbitopathy, cystinosis, and neurofibromatosis.

### 2.3. Examinations

The clinical examination of each patient was divided into a history and a physical examination.

#### 2.3.1. Medical History

The medical history of an individual patient included their family history; the presence of disease symptoms at birth and during life; the type and distribution of scaling and scalp, eyebrow, and lash abnormalities; the involvement of extremities; and extracutaneous symptoms. The type of ichthyosis was assessed based on the results of the dermatological consultation provided by the patients, which was supported by the genetic testing analysis in two out of five patients (P1, P5). 

#### 2.3.2. Ocular Surface Assessment

Imaging of the anterior segment of the eye was performed using a slit-lamp biomicroscope (SL 9900; Haag-Streit Type, CSO, Florence, Italy). Both eyes were photographed at magnitudes of 10× and 16×. Ocular surface tests were performed, including tear break-up time (TBUT) and eye surface staining, conducted using a fluorescein strip (Fluoro touch; 1 mg fluorescein sodium; Madhu Instruments, Delhi, India). 

TBUT stands for the measured time in seconds between the full opening of the eyelids after two complete blinks and the first break in the tear film observed by an examiner. The Oxford scheme for grading ocular surface staining was used [28]. Patients with TBUT < 10 s or ocular surface staining result of >5 points for defects of the corneal epithelium or >9 for defects of the conjunctiva or epitheliopathy of the eyelid margin or = 2 mm long and/or = 25% width were classified as abnormal. 

#### 2.3.3. SS-OCT

Corneal tomographic measurements and morphological assessments were obtained via SS-OCT, with a CASIA 2 (Tomey Corporation, Inc., Nagoya, Japan) with an infrared light wavelength of 1310 nm. The examination was performed in a naturally ventilated room with no glare sources. All measurements were acquired using the automated mode: auto-alignment and auto-shot functions. The measurements with the quality factor marked as “OK” were used for further analysis. Anterior segment imaging was performed by one observer (A.M.) and additionally validated by a second (A.N.). The “Corneal Map” protocol (16 radial scans; 800 A-scans per line) was used to evaluate the topography and pachymetry data. The following parameters regarding keratometric (k), posterior (p), and real (r) topographic data were evaluated: flat, steep keratometry (Kf, Ks), astigmatism (CYL), average keratometry (AvgK), and average central corneal power (ACCP). Additionally, the area analyzed (AA, %) and the corneal curve (Ecc)’s eccentricity were assessed. AA reflects the percentage of the performed automatic analysis at a 10 mm diameter area. Ecc corresponds to the numeric value of the eccentricity at a 9 mm diameter area. The pachymetry map parameters included the corneal thickness at the apex (CAT; um) and the corneal thinnest thickness (CTT; um). The anterior chamber depth (ACD; mm) was measured.

Fourier indices were automatically derived from the keratometric (k), anterior (a), and posterior (p) topographic maps in the diameter range of 3–6 mm. The assessed parameters included the following components: spherical (zero order), asymmetric (first order), regular astigmatism (second order), and irregular astigmatism (higher order). 

Fourier analysis of the axial power data (keratometric and posterior) was used to evaluate the Ectasia Screening Index (ESI), which is a parameter used to detect anterior and posterior corneal ectasia-specific patterns. The ESI (%) is assessed based on the following data: the thinnest corneal thickness (μm) and its location relative to the corneal apex (mm) (X, Y coordinates); the FI of the keratometric and posterior topography data; and the location and the lowest value of the instantaneous posterior power within a 6 mm diameter. The results were color coded, with red representing 30% or higher (ectasia), yellow representing the range from 5 to 29% (suspected ectasia pattern), and green for less than 4% (pattern within the normal range). A detailed description of each specific SS OCT parameter was described in our previous paper and is also presented in Appendix A [25].

#### 2.3.4. IVCM

The contact Rostock Cornea Module of the Heidelberg Retina Tomograph (Heidelberg Engineering GmbH, Dossenheim, Germany) was used to obtain confocal images of each representative corneal layer. Anesthetic eye drops of 0.5% proparacaine hydrochloride (Alcaine, Alcon Laboratories, Fort Worth, TX, USA) and an ophthalmic gel medium (Vidisic eye gel, Bausch and Lomb, Berlin, Germany) were used to prepare eyes for the examination. This exam was the last one performed, after the ocular surface assessment and OCT, due to its invasive character. The central part of the cornea was assessed (3–4 mm diameter), the mean examination time was 5 min per eye, and the mean number of acquired images was 15 scans per eye. The examination was performed by one observer (J.K.L.) and the validation of the results was confirmed by the second observer (A.N.). All scans were analyzed and the representative scans presenting abnormalities were chosen.

## 3. Results

The study group consisted of five patients (P1–P5) aged 13–66 years (three females and two males). The BCVA (best corrected visual acuity; decimal scale) ranged from light perception (LP) to 1.0 (median 0.25). Additionally, genetic analysis results were obtained from six family members of patients P1 and P5 (aged 10–58 years; four females and two males), and slit-lamp examination results were obtained from five family members (aged 10–58 years; three females and two males).

### 3.1. General Demographic Information and Medical History

A summary of the demographic and general medical history data is presented in Table 1.

The family history was positive in P1, who was diagnosed with LI, and P4, who was diagnosed with HI. Patient P1’s female sibling was also affected but was not included in the study group (consent not given). Patient P4’s male sibling died shortly after birth because of ichthyotic complications. Otherwise, no diagnosis of ichthyosis was made in any family member in the study group. 

ARCI was diagnosed based on a dermatological consultation in all patients except P3, who was diagnosed with IV (P1, P2–LI, P4–HI, P5–BSI, LI minor variant).

Two patients received genetic confirmation of the diagnoses (P1 and P5), and genetic analysis results were obtained from six family members. Next-generation sequencing (NGS) was performed using a MiSeq sequencer (Illumina, San Diego, CA, USA) with the SeqCap EZ HyperCap protocol and the NimbleGen SeqCap EZ probe kit (Roche Sequencing Solutions, Inc., Basel, Switzerland). NGS analyseshad a mean read coverage of 121.1×. The results were confirmed by Sanger sequencing. The Human Genome Variation Society nomenclature (HGVS v15,11) was used to describe the revealed mutations. To analyze TGM1, a reference sequence with accession number NM_000359.3 (HGMD) was used. The known homozygous, pathogenic variant c.[943C>T] (p.Arg315Cys) was found following TGM1 analysis in P5 (proband). The TGM1 sequencing of both parents and the proband’s sister revealed the same variant in a heterozygous state. The known heterogeneous mutation c.579G>A (p.Trp193Ter) on one allele and c.1135G>C (p.Val379Leu) on the second allele of the TGM1 gene was found in patient P1 (proband). Patient P1’s sister (18 years old), with the LI phenotype, presented with identical genetic results. Moreover, P1’s mother was identified as having the mutation p.Trp193Ter in one allele, and P1’s father was identified as having p.Val379Leu in one allele of the TGM1 gene. 

### 3.2. Slit-Lamp Findings

Severe involvement of the eyelids with cicatricial ectropion of the upper and lower eyelids was present in two patients (P2 and P4) diagnosed with LI and HI, respectively. P1, diagnosed with LI, had ectropion in the lower eyelids, whereas the upper eyelids were spared. The eyelids of P3, who was diagnosed with IV, and those of P5, diagnosed with BSI, had normal anatomy. All patients presented with the hyperkeratinization of the lid margin, with scaling and crusting at the base of the lashes to different degrees of severity (Figure 1). Patients P1, P2, and P4 were severely affected, with keratinization crossing the lid margin towards the conjunctival fornices. P3 exhibited moderate involvement compared to P1, P2, and P4, while scaling and keratinization were mild in P5.

All patients were diagnosed with moderate-to-severe DED. Abnormal TBUT and fluorescein staining were observed in all eyes. A summary of the TBUT and fluorescein staining results is presented in Table 2. Exposure keratopathy with corneal scarring and peripheral vascularization of different levels of severity due to ectropion was diagnosed in patients P1, P2, and P4. P2, with bilateral ectropion, severe eyelid deformation, and exposure keratopathy, was diagnosed with a diffuse corneal scar after corneal perforation in the temporal paracentral quadrant, anterior iris synechiae, shallow anterior eye chamber, and cataract (Figure 1B,C). Notably, corneal scarring with peripheral vascularization and limbal stem cell deficiency was present not only in patients diagnosed with ectropion but also in P3 (Figure 1D,E). In addition, lipid keratopathy with corneal vascularization and calcification was noted in P4 (Figure 1F).

Representative slit-lamp photographs are shown in Figure 1. A slit-lamp examination of five family members of the patient did not reveal any abnormalities.

### 3.3. Optical Coherence Tomography

#### 3.3.1. Corneal Topography and Pachymetry Results

The detailed results of the corneal parameters are presented in Appendix A, and the summarized results are presented in Table 2. Three patients, P2 (LI), P3 (IV), and P4 (HI), were diagnosed with ectasia patterns based on ESI. The results ranged from 7 to 95% of the probability of a corneal ectasia pattern. The CAT ranged from 480 to 734 µm. The corneas showed a significant pattern of irregularity, with a large difference between CAT and CTT exceeding 375 µm. The thinnest point displacement from the inferotemporal region was observed in P2.

The results of the corneal shape parameters of the left eye of P2 and the right eye of P3, which were severely affected (Figure 1C,D), were significantly different in terms of keratometric, posterior, and real keratometry readings, as well as the cylinder results. Both corneas were very steep (kAvgK 66.2 and 55.4, respectively) with a high degree of keratometric astigmatism (kCYL 6.2 and 3.8, respectively). Figure 2 shows the ESI results of severely affected eyes with ectasia patterns in P2 and P3.

#### 3.3.2. Fourier Indices

The detailed FI results for all the parameters are presented in Appendix A, and the summarized results are presented in Table 3. All patients revealed abnormalities in at least one parameter, namely, regular astigmatism for at least one eye at a diameter of 3 or 6 mm for keratometric, anterior, or posterior analysis.

The FI parameters that showed abnormalities in at least one eye were primarily located on the posterior corneal surface and included 3 mm p reg, astigmatism, 3 mm p asymmetry, 3 mm p higher order, 6 mm p reg, astigmatism, and 6 mm p higher order. Patients P2, P3, and P4 had abnormalities in almost all anterior, keratometric, and posterior FIs, which aligned with the ectasia corneal pattern in the topography results (Figure 2).

#### 3.3.3. Morphology

Most morphological OCT findings are related to severe DED, limbal stem cell deficiency, exposure keratopathy, scarring, and corneal surface irregularity. Scar tissue was visible as an irregular area of hyperreflectivity in the anterior stroma. Scarring was evident in all patients except P5. The most significant corneal changes due to keratopathy exposure were observed in the paracentral lower corneal quadrant. Local irregularities in the anterior and posterior corneal surfaces were also visible in all patients, except for P5. Anterior iris synechiae were present in P2 and P3. Local (P3) or generalized (P2) corneal edema was present in two patients. Representative images of the OCT scans are shown in Figure 3.

### 3.4. In Vivo Confocal Microscopy

Representative IVCM images of each patient are shown in Figure 4. The epithelium showed different degrees of abnormality, including epithelial squamous metaplasia, an irregular shape, and hyperreflectivity patterns of cells with hyperreflective patches. The nerve plexus only showed normal anatomy in both eyes of P5 and the RE of P4. Otherwise, it was completely or partially invisible because of corneal scarring related to exposure keratopathy. Figure 4C shows a tortuous nerve plexus with decreased nerve density and haze present at the depth of the Bowman layer in P3. Stromal changes in patients P1–P5 ranged in severity and form and mostly comprised keratocyte pleomorphism, enlarged hyperreflective activated keratocytes, stromal haze, scarring, hyperreflective opacities, vascularization, crystalline lipid keratopathy, and stromal hyperreflective folds. The least-affected individual, P5, showed only corneal stromal abnormalities in the form of multiple stromal microdot deposits within the stroma. The endothelial layer also showed abnormalities (P1, P3, and P4) or was impossible to visualize because of the extensive scarring (P2). Hyperreflective opacities and pleomorphism were also observed.

## 4. Discussion

Although ocular involvement may be regarded as a significant feature of ichthyoses and includes multiple symptoms, based on slit-lamp examinations, it usually presents in relatively small patient sample sizes, or as case reports.

In this study, we aimed to define the multimodal characteristic features of the ocular surface in patients with ichthyosis. We evaluated the results of the slit-lamp examination, anterior segment SS-OCT, and IVCM in patients with IV and ARCI, including HI, LI, and BSI. To date, only a few reports have included detailed objective ocular surface assessments based on novel imaging techniques such as meibography, optical coherence tomography, Scheimpflug imaging, and confocal microscopy [18,29,30,31]. Multimodal imaging is mostly available for specific syndromes, including keratitis–ichthyosis–deafness (KID), ichthyosis follicularis, alopecia, photophobia syndrome (IFAP), Sjögren–Larsson syndrome (SLS), or pre-Descemet’s membrane corneal dystrophy (PDCD) [8,9,10,11,32,33,34,35,36,37]. 

Based on our results, several significant ocular surface abnormalities were identified, as is consistent with previous research; however, the study also provided new data, specifically related to ectasia patterns and Fourier index aberrations based on SS-OCT, and insights into microscopic structural changes revealed by IVCM. 

The corneal epithelium plays a crucial role in maintaining ocular surface homeostasis and is an important barrier to pathogens and environmental agents. The epithelial abnormalities in our group included epithelial squamous metaplasia, irregular shapes, and cells with hyperreflective patches. These features may be compared to those of patients with severe dry eyes, including epithelial squamous metaplasia with an increase in desquamation, enlarged cells, pyknotic nuclei, and a lower cell density than normal. In addition, several abnormalities related to the nerve plexus were observed in the study group. The nerve plexus showed normal anatomy only in P5 and in the RE of P4. Otherwise, it was completely or partially invisible because of corneal scarring, which is mostly related to exposure keratopathy. Reductions in sub-basal nerves have also been described in congenital diseases, such as congenital corneal anesthesia (CCA) and corneal dystrophies, and acquired diseases, such as dry eye, diabetes, and infectious keratitis, as well as herpes simplex, and bacterial, fungal, and Acanthamoeba keratitis [38,39,40,41]. In ichthyoses, nerve plexus abnormalities may be caused by dry eyes, keratopathy, excessive scarring, and limbal stem cell deficiency. Corneal vascularization is secondary to keratopathy and the underlying limbal stem cell deficiency. Notably, corneal scarring with peripheral vascularization and local or diffuse limbal stem cell deficiency was present not only in patients diagnosed with ectropion but also in P3 (IV, normal eyelid anatomy). All patients were characterized by stromal involvement, ranging from the mildest stromal microdot deposits (P5) to severe scarring (P2). However, the origin of the microdot deposits remains unclear. One theory is that the microdots consist of lipofuscin granules of intracellular origin. They have been reported in healthy subjects and are regarded as early-stage irreversible corneal stromal alterations due to hypoxia [42]. Utheim et al. reported an accumulation of microdots in the aging cornea, primarily in the subepithelial stromal region [43]. This conclusion does not align with our findings, since P5 is young (13 years old) and presented with microdot accumulation in the anterior and posterior stroma. 

To assess the anterior and posterior corneal surface disturbances, we used anterior segment SS-OCT. To date, no published data are available regarding the SS-OCT findings in patients with ichthyosis. However, a study on the topographic and biomechanical evaluation of the cornea in patients with ichthyosis vulgaris was conducted by Kara, N., et al. [29]. There was another reported case of marginal pellucid degeneration in a patient with IV disease (44). Moreover, based on Scheimpflug imaging, Palamar, M., et al. detected bilateral keratoconus in 2 out of 12 patients with genetically confirmed LI (19). Kara, N., et al. revealed that, while the corneal topographic findings and corneal hysteresis (CH) in patients with IV were similar to those in healthy subjects, the mean corneal resistance factor (CRF) and central corneal thickness (CCT) were significantly lower in patients with IV [29]. These results conflict with our findings because, in the case of patient P3, diagnosed with IV, we revealed a significant corneal ectasia pattern in both eyes (CTT, 356 µm and 459 µm; CAT-CTT, 217 µm and 93 µm; ESI, 95% and 50%, respectively, for RE and LE; Appendix A and Table 2). It should be emphasized that ichthyosis vulgaris was strongly associated with atopy, which may have resulted in the diagnosis of ectatic corneal disorders in our patient. Bilateral ectasia patterns were observed in patients P2 (LI) and P4 (HI). We demonstrated an ectasia profile in five of ten eyes with various ichthyotic forms, including IV, LI, and HI. The high percentage of ectopic patterns in our study group may be explained by the high sensitivity and specificity of SS-OCT in diagnosing and grading keratoconus compared to the slit-lamp examination used in other studies [23,24,25]. A recent Scheimpflug imaging study conducted by Palamar, M., et al. suggested that the rate of keratoconus is underestimated in patients with ichthyosis and that all patients should undergo topographic screening for ectatic disorders [18]. Although the imaging devices of Scheimpflug topography and SS-OCT rely on different principles, both techniques are characterized by the high reliability of the corneal parameter measurements. On this basis, one can conclude that the rate of keratoconus is significantly higher in patients with ichthyosis than in the general population (approximately 1.38 per 1000 people) [44]. 

Our study demonstrated that all patients revealed abnormalities in at least one parameter of the Fourier indices, namely, regular astigmatism for at least one eye of 3 or 6 mm diameter for keratometric, anterior, or posterior analysis. The high incidence of FI abnormalities can be explained by the fact that the results may be compromised by subclinical, non-specific abnormalities of the corneal surface, and by the influence of other factors, such as environmental factors and tear film instability. Abnormalities in the tear film have previously been reported in ichthyosis and were confirmed by our abnormal TBUT and fluorescein staining results [17,30]. All the patients from our study group were diagnosed with DED. Subsequently, the punctate keratitis may be due to the tear film instability caused by MGD or secondary cicatricial eyelid margin contraction. Corneal exposure often leads to ulcers, perforations, and severe scarring. Based on a series of 10 patients with LI and cicatricial lagophthalmos published by Cruz, A., et al., 30% developed corneal exposure leading to the loss of useful vision [45]. Our results are similar because exposure keratopathy with corneal scarring and peripheral vascularization of different severities due to ectropion was diagnosed in three patients (P1, P2, and P4). The median BCVA was 0,25, ranging from LP to 1,0. The severely affected P2 with LI was diagnosed as having a diffuse corneal scar after a corneal perforation in the temporal paracentral quadrant (Figure 1C). Notably, corneal scarring with peripheral vascularization and limbal stem cell deficiency was present not only in patients diagnosed with ectropion, but also in P3, who had IV and normal eyelid anatomy (Figure 1D,E). 

This study has certain limitations. The main limitation is the small sample size; five patients were diagnosed with different forms of ichthyosis: IV, LI, HI, and BSI. A larger series is needed to confirm our results concerning the SS-OCT and IVCM findings. Additionally, this was not a screening study. The study group consisted of patients referred to an ophthalmologist. Therefore, the severity of the ophthalmic findings may be overestimated when compared with the entire patient population. There was a high diversity of ophthalmic findings, ranging from a preclinical state (P5) to severe involvement (P1, P2, P4). Moreover, providing a serial assessment would be valuable, especially one focused on a long-term observation period. The observation time was relatively short and lasted for 12 months. Moreover, no reliable statistical analyses were possible for this small and diverse study group. The repeatability and reproducibility of SS-OCT have been proven in healthy corneas; however, studies with larger samples of unhealthy eyes are lacking. To the best of our knowledge, this is the first observational study to assess a wide range of corneal parameters based on SS-OCT in ichthyosis. 

It is also worth noting that genetic counseling plays a crucial role in the final diagnosis and might enable physicians to predict the possibility of upcoming ocular problems in patients with ichthyosis. Two patients in our study group had genetically confirmed ARCI. *TGM1* gene mutations have been identified. Mutations in this gene are the predominant cause of ARCI, particularly the LI subtype. The known homozygous pathogenic variant c.[943C>T]; (p.Arg315Cys) was found following TGM1 gene analysis in P5 (proband), and was strongly linked to BSI [14,46]. The suggested effect of p.Arg315Cys on TGase-1 function includes low specific activity, presumably because of protein misfolding or excessively stable proteins that cannot be processed. The other three patients were diagnosed based on dermatological consultation results, which also occur in most other ichthyosis studies [16,17,19,20,29]. 

In summary, our multimodal study revealed several characteristic ocular surface features that may be overlooked when using slit-lamp examinations alone. These early changes include DED, punctate keratopathy, and peripheral vascularization, which may occur regardless of anatomical eyelid disturbances. Therefore, patients with ichthyosis should be examined and treated for the early onset of DED. We also revealed that IVCM might aid in assessing the severity of ocular surface diseases and lead to an improved understanding of the pathophysiological mechanisms of this complex disease. Furthermore, owing to the visualization of the subclinical findings, IVCM may allow for the detection of ocular complications at much earlier stages. Our study also showed that SS-OCT could detect anterior and posterior corneal surface abnormalities and reveal subclinical ectatic patterns. Screening ichthyosis patients for possible keratoconus is a significant finding that should be acknowledged. Moreover, SS-OCT allows for the evaluation of the distribution of corneal opacities and illustrates the depth of scarring and shape abnormalities. 

## 5. Conclusions

Combining the multimodal imaging results, we detected preclinical abnormalities, distinguished characteristic changes common to ichthyosis, and revealed the depth and nature of the corneal abnormalities.

Further studies may reveal whether it is beneficial for patients for physicians to predict the outcome of their disease at an early stage and to provide sufficient ocular-surface-microstructure-based treatment. 

## Figures and Tables

**Figure 1 jcm-12-06006-f001:**
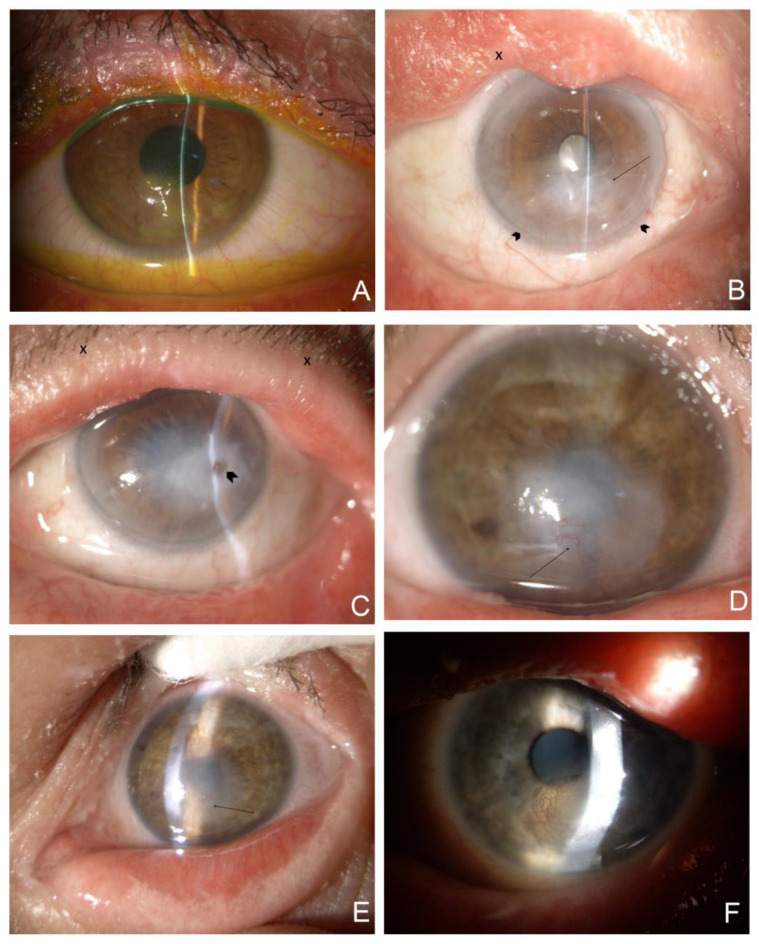
Representative slit-lamp eye photographs of the study group. The quality of the photos is compromised due to the increased light sensitivity among ichthyosis patients. (**A**) (mag. 10×, after the installation of fluorescein) P1. RE. (LI) Excessive scaling with hyperkeratinization of the lid margin and crust and scales at the base of the eyelashes. Severe exposure keratopathy with significant irregular fluorescein staining located in the lower half of the cornea. (**B**) (mag. 10×) P2. RE. (LI) Cicatricial ectropion of the upper and lower eyelids with eyelid margin deformation and hyperkeratinization (X), arcus lipoides, peripheral vascularization on two thirds of the lower cornea (arrowheads), corneal scarring with irregularity of the epithelium due to exposure keratopathy (arrow). (**C**) (mag. 10×) P2. LE. (LI) Cicatricial ectropion of the upper and lower eyelids with eyelid margin hyperkeratinization and crust at the base of the eyelashes (X); arcus lipoides with mild corneal peripheral vascularization, diffuse, dense, central, and paracentral corneal scars with temporal focal pigmentation subsequent to corneal perforation (arrowhead). (**D**) (mag. 16×) P3. RE. (IV) Localized, central irregular scar involving the lower peripheral cornea. Invading vessels from the lower periphery (arrow). (**E**) (mag. 10×) P3. LE. (IV) Hiperkeratinisation of the lower lid exceeding the lid margin, central, diffuse corneal scar with vessel ingrowth from the lower cornea (arrow). (**F**) (mag. 10×) P4. LE. (HI) Upper eyelid deformation. Paracentral area of reticular vascularization, lipid keratopathy, and calcification.

**Figure 2 jcm-12-06006-f002:**
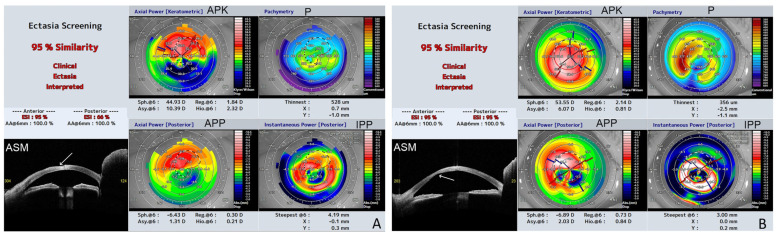
Representative results of swept-source OCT (SS-OCT); the Ectasia Screening Index (ESI) report of the affected eyes from the study group (a, anterior; p, posterior). The report includes the following data: APK (axial power keratometric); APP (axial power posterior); Sph.6: spherical component of FI at a 6 mm diameter (D), Reg.6: regular astigmatism component at a 6 mm diameter (D), Asy.6: asymmetric component at a 6 mm diameter (D), and Hio.6: higher-order irregular astigmatism component at a 6 mm diameter (D); P (pachymetry), the corneal thickness (μm) of the thinnest part and the location relative to the corneal apex (coordinates: X, Y) (mm); IPP (instantaneous power posterior), Steepest@6 (mm): the value of the steepest instantaneous posterior power and its location relative to the corneal apex (coordinates: X, Y) (mm). ASM (anterior eye segment morphology) line scan. (**A**) P2, RE (LI). Note the high degree of irregularity on the anterior corneal surface (arrow). The 6 mm k Asymmetry FI is 10.39 D, which reflects the significant difference between the upper and lower anterior corneal surfaces. The summarized similarity of the ectasia corneal pattern is 95% (aESI 95%; pESI 66%). (**B**) P3, RE (IV). Note the high degree of irregularity on the posterior corneal surface (arrow); the CTT decreased to 356 µm, and Sph@6 increased to 53.55 D. The summarized similarity of the ectasia corneal pattern is 95% (aESI 95%; pESI 95%).

**Figure 3 jcm-12-06006-f003:**
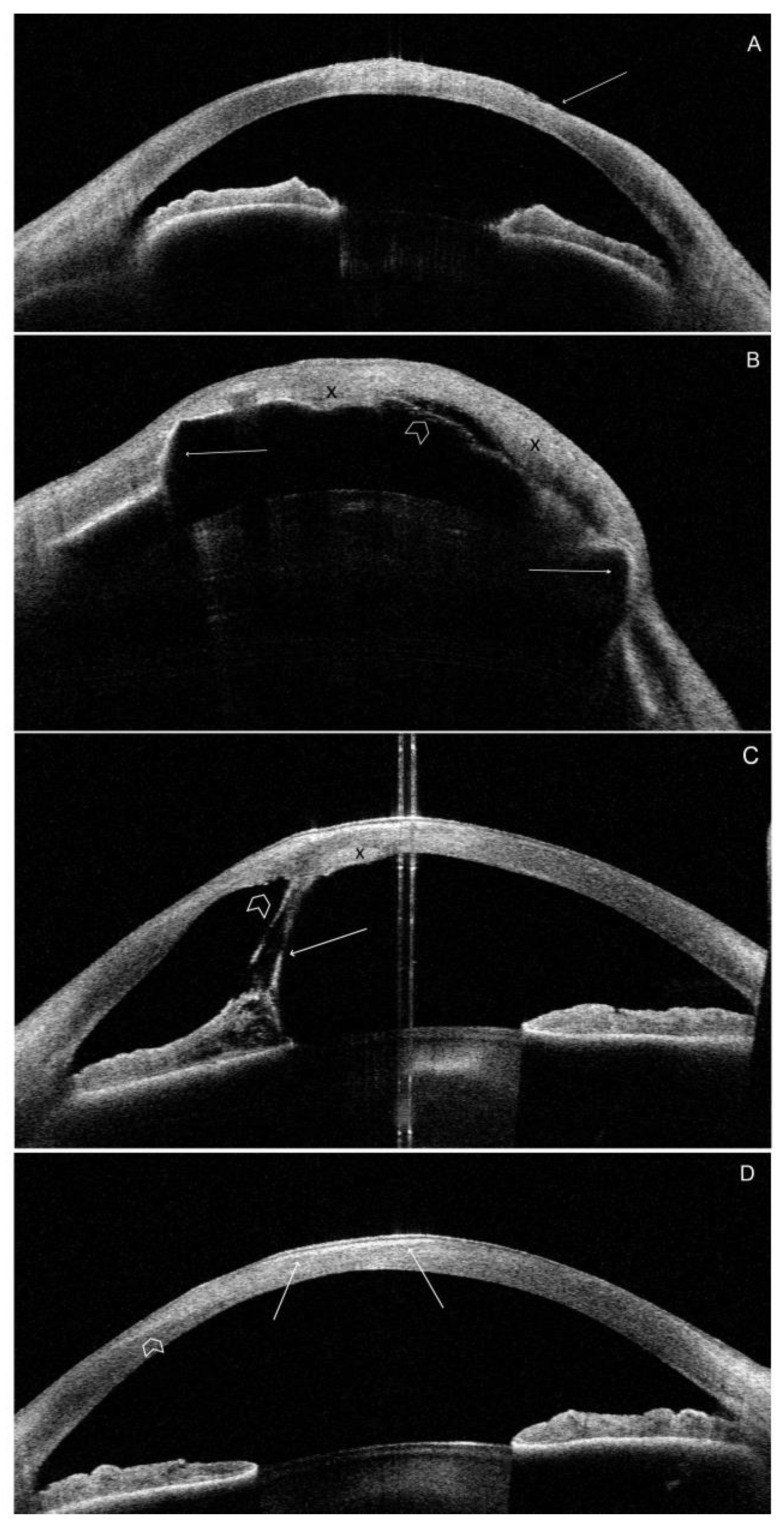
Representative high-definition morphology SS-OCT scans of the most severely affected patients. (**A**) P2. RE. (LI). Line 0–180°. Irregularity of the anterior corneal surface (arrow). Increased reflectivity of the corneal stroma corresponding with the area of the corneal scar (Figure 1B). (**B**) P2. LE. (LI). Line 0–180°. Generalized anterior iris synechiae (arrows). Significant irregularity on both corneal surfaces. Increased reflectivity of the stroma due to diffuse, dense corneal scar subsequent to exposure keratopathy (Figure 1C). Corneal edema with different local severity (X). The area of Descemet membrane detachment, shown paracentrally (arrowhead). Increased lens reflectivity due to cataract. (**C**) P3. RE. (IV). Line 80–260°. Irregular anterior and posterior corneal surface. Central hyperreflectivity corresponding to the central corneal scar visible in Figure 1D. Anterior iris synechiae (arrow). The localized Descemet membrane rupture (arrowhead) and local corneal edema (X). (**D**) P4. LE. (HI). Line 150–330°. Marked hyperreflectivity in the anterior and central corneal sections (arrows). Additionally, the diffuse region of hyperreflectivity extends to the paracentral part at the axis of 330° (arrowheads). This region of increased reflectivity corresponds to the central lipid keratopathy with a reticular pattern of vascularization (Figure 1F).

**Figure 4 jcm-12-06006-f004:**
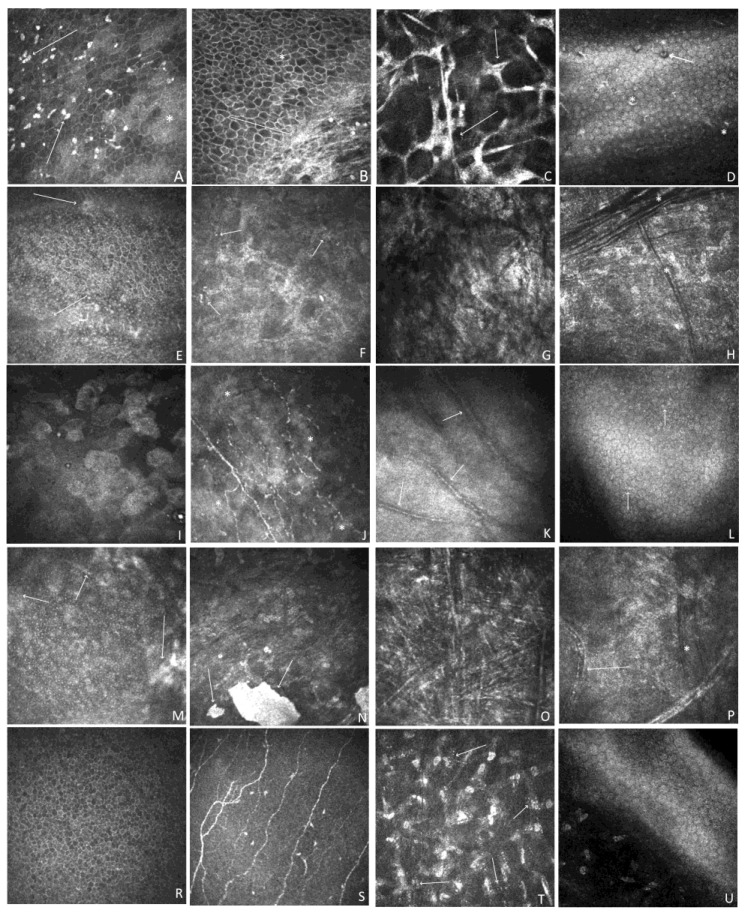
Representative IVCM images at different scanning depths. (**A**–**D**); P1. RE. (**A**) Confocal image at a depth of 21 µm. Corneal epithelium. The irregular shape of cells, and hyperreflectivity pattern of cells with hyperreflective patches (*), along with multiple highly reflective irregular deposits (arrow). (**B**) Confocal image at a depth of 55 µm. Irregularity of cells with hyperreflective patches (*), subepithelial fibrosis with an absent nerve plexus (arrow). (**C**) Confocal image at a depth of 330 µm. Stroma. Confluent group of abnormally hyperreflective keratocyte nuclei with visible cytoplasmic processes (arrow). (**D**) Confocal image at a depth of 530 µm. Endothelium. Hyperreflective, small precipitates (*), a few hyporeflective spots containing a central highlight (arrow). (**E**–**H**); P2. RE. (**E**) Confocal image at a depth of 48 µm. Corneal epithelium. Marked irregularity of cells. Notable hyperreflective patches (arrow). (**F**) Confocal image at a depth of 175 µm. Stromal haze and scaring. Numerous small hyperreflective dots (arrow). (**G**) Confocal image at a depth of 300 µm. Non-homogenous area of massive scarring and fibrosis. (**H**) Confocal image at a depth of 400 µm. Dark differently orientated striae (*). Stromal haze. Keratocyte nuclei are barely distinguishable. (**I**–**L**); P3. RE. (**I**) Confocal image at a depth of 15 µm. Epithelial squamous metaplasia. Enlarged, hyperreflective cells, irregularly arranged, and decreased cell density. (**J**) Confocal image at a depth of 54 µm. Loss of nerves increased tortuosity. Hyperreflective, non-homogenous patches (*). (**K**) Confocal image at a depth of 320 µm. Hyperreflective scarring with neovascularization (arrow). (**L**) Confocal image at a depth of 566 µm. Cell pleomorphism. Hyperreflective, small precipitates (arrow). (**M**–**P**) P4. LE. (**M**) Confocal image at a depth of 30 µm. Marked irregularity of cells. Hyperreflective areas covering epithelium (arrow). (**N**) Confocal image at a depth of 80 µm. Homogenous, hyperreflective, distinctive area of calcification (arrow). Stromal haze and keratocyte nuclei are barely distinguishable. Multiple, small microdots. (**O**) Confocal image at a depth of 310 µm. Multiple hyperreflective needle-like opacities, which are differently orientated. Crystalline lipid keratopathy (**P**) Confocal image at a depth of 520 µm. Scaring with vessels (arrow). Dark stromal striae (*). (**R**–**U**) P5. LE. (**R**) Confocal image at a depth of 20 µm. Corneal epithelium. (**S**) Confocal image at a depth of 60 µm. Nerve-plexus. (**T**) Confocal image at a depth of 340 µm. Multiple stromal microdots (arrow). (**U**) Confocal image at a depth of 550 µm. Posterior stroma and endothelium. Image quality is compromised due to the patient’s age and poor cooperation during the examination.

**Table 1 jcm-12-06006-t001:** Demographic data and medical history data of the study group.

	Patient 1 (P1)	Patient 2 (P2)	Patient 3 (P3)	Patient 4 (P4)	Patient 5 (P5)
Age	20	66	44	23	13
Gender	Male	Female	Female	Female	Male
Family history	Female sibling affected			A male sibling died shortly after birth due to an ichthyosis complication	
Scaling at birth	Present	Present	No	Present	Present
Type and distribution of scaling	Large, rhomboid, dark brown; generalized	Large, light. Brownish; generalized	Small, white, and gray; generalized	Coarse and platelike, white; generalized	Small white, grey on the face, brownish on the trunk; the disparity between trunk and extremities and face
Extremities	Skin deprived of sweat glands, hairless;	Skin deprived of sweat glands, hairless;	Sweat glands present; Skin hairless;	Synechiae of digits; skin deprived of sweat glands, hairless;	Excessive scale in armpits, in the bends of the elbows and knees; sweat glands and hair present
Scalp abnormalities, eyebrows, lashes	Scarring alopecia, no eyebrows, lashes on the upper and lower eyelid present	Localized temporal alopecia, otherwise hair on the head, brows, and lashes on the upper eyelid present	Hair on the head, brows, and lashes on the upper and lower eyelid present	Synchiae of auricles, scarring alopecia, brows, and lashes on the upper eyelid present, but very brittle	Hair on head, eyebrows, and lashes present, but fair and brittle
Symptoms general	Cryptorchidism at birth, Hypercholesterolemia	The short statue, failure to thrive, Hypercholesterolemia	None	Hearing affected, short statue, failure to thrive	Sepsis at birth, intraventricular leakage of open foramen ovale
Type of ichthyosis	ARCI LI (genetic confirmation)	ARCI LI	IV	ARCI HI	ARCI BSI (LI minor variant) (genetic confirmation)

ARCI, autosomal recessive congenital ichthyosis; LI, lamellar ichthyosis; IV, ichthyosis vulgaris; BSI, bathing suit ichthyosis; HI, harlequin ichthyosis.

**Table 2 jcm-12-06006-t002:** Summarized BCVA, ocular surface examination, and corneal topography and thickness map results. The detailed results are presented in Appendix A.

Parameter	P1	P2	P3	P4	P5	Min	Max	Median
BCVA (decimal)	RE 0.8	RE 0.2	RE 0.05	RE 0.3	RE 1.0	LP	1.0	0.25
	LE 0.7	LE LP	LE 0.1	LE 0.2	LE 0.4			
TBUT (s)	RE 6	RE 4	RE 7	RE 5	RE 12	4	10	6.5
	LE 8	LE 4	LE 8	LE 5	LE 8			
Fluorescein staining (Oxford scale)	RE IV	RE III	RE IV	RE II	RE 0	0	IV	2.0
	LE II	LE IV	LE III	LE II	LE I			
kAvgK (D)	RE 42.1	RE 47.0	RE 55.4	RE 48.9	RE 43.9	41.8	66.2	47
	LE 41.8	LE 66.2	LE 49.8	LE 47.0	LE 43.5			
pAvgK (D)	RE −6.0	RE −6.5	RE −7.6	RE −7.4	RE −6.4	−10.1	−6	−6.5
	LE −6.2	LE −10.1	LE −6.5	LE −6.7	LE −6.3			
rAvgK (D)	RE 41.0	RE 46.1	RE 55.3	RE 47.3	RE 42.6	40.5	64.4	45.9
	LE 40.5	LE 64.4	LE 49.2	LE 45.7	LE 42.2			
rCYL (D)	RE 2.0	RE 2.4	RE 5.4	RE 2.6	RE 2.1	0.8	7.3	2.1
	LE 0.8	LE 1.9	LE 7.3	LE 1.0	LE 2.1			
CAT (µm)	RE 539	RE 606	RE 573	RE 480	RE 545	480	734	546.5
	LE 529	LE 734	LE 552	LE 505	LE 548			
CTT (µm)	RE 520	RE 528	RE 356	RE 462	RE 540	356	540	503.5
	LE 509	LE 395	LE 459	LE 498	LE 540			
CAT-CTT (µm)	RE 10	RE 78	RE 217	RE 18	RE 5	5	375	19
	LE 20	LE 375	LE 93	LE 7	LE 8			
ACD (mm)	RE 2.97	RE 2.27	RE 2.86	RE 3.03	RE 3.02	1.9	3.03	2925
	LE 2.94	LE 1.9	LE 2.81	LE 2.95	LE 2.91			
ESI (%)	RE 0	RE 95	RE 95	RE 61	RE 0	0	95	28.5
	LE 0	LE 95	LE 50	LE 7	LE 0			

**Table 3 jcm-12-06006-t003:** Summarized results of Fourier Indices presenting parameters that showed abnormalities in at least four patients. The detailed results are presented in Appendix A.

Parameter	P1	P2	P3	P4	P5	Min	Max	Median
6 mm k Reg. Astigmatism	RE 1.08 *	RE 1.84 *	RE 2.14 *	RE 1.48 *	RE 1.21 *	0.3	4.86	1.39
	LE 0.3	LE 4.86 *	LE 3.32 *	LE 0.6	LE 1.31 *			
6 mm a Reg. Astigmatism	RE 1.2 *	RE 2.06 *	RE 2.38 *	RE 1.64 *	RE 1.35 *	0.33	5.41	1.55
	LE 0.33	LE 5.41 *	LE 3.69 *	LE 0.67	LE 1.46 *			
3 mm p Reg. Astigmatism	RE 0.14	RE 0.38 *	RE 0.88 *	RE 0.34 *	RE 0.35 *	0.14	1.93	0.34
	LE 0.14	LE 1.93 *	LE 0.23	LE 0.19	LE 0.33			
3 mm p Asymmetry	RE 0.12	RE 1.25 *	RE 1.97 *	RE 1.09 *	RE 0.07	0.03	1.97	0.15
	LE 0.15 *	LE 0.41 *	LE 0.86 *	LE 0.14 *	LE 0.03			
3 mm p Higher Order	RE 0.04*	RE 0.23 *	RE 0.93 *	RE 0.07 *	RE 0.02	0.02	1.81	0.06
	LE 0.06 *	LE 1.81 *	LE 0.16 *	LE 0.03	LE 0.03			
6 mm p Reg. Astigmatism	RE 0.13	RE 0.3	RE 0.73 *	RE 0.34 *	RE 0.32 *	0.12	1.74	0.31
	LE 0.12	LE 1.74 *	LE 0.29	LE 0.18	LE 0.31 *			
6 mm p Higher Order	RE 0.05	RE 0.21 *	RE 0.84 *	RE 0.06 *	RE 0.02	0.02	1.81	0.06
	LE 0.06 *	LE 1.81 *	LE 0.15 *	LE 0.03	LE 0.04			

* out of reference range (normative database); RE, right eye; LE, left eye; (k), keratometric; (a) anterior; (p) posterior; reg., regular.

## Data Availability

The data presented in this study are available on request from the corresponding author. The data are not publicly available due to the Ethical Committee’s indication.

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
