# Peer review of "Advanced Anterior Eye Segment Imaging for Ichthyosis"

_jcm, 2023, doi:10.3390/jcm12186006_

Round 1

Reviewer 1 Report

The purpose of this study was to describe ocular surface and anterior eye segment findings in various types of ichthyoses and to analyze the results of multimodal imaging (ssOCT, IVCM; single-center prospective observational study; five patients with different ichthyosis forms) and ocular surface assessment. The authors were able to detect preclinical abnormalities, and to distinguish characteristic changes common to ichthyosis. Patients with ichthyosis should be examined for dry eye disease and ectatic disorders (keratoconus) early in clinical practice. The partly new results (ssOCT of corneal parameters) provide a profound understanding of the features associated with different variants of ichthyosis. Limitations: small sample size, no longitudinal data. Content is of high clinical value although the disease occurs very rarely. Great discussion, very instructive Figures. Congratulations.

Author Response

We appreciate your comments and clues. We strongly agree, that due to its rarity ichthyosis and its ocular complications should be disused to raise awareness among ophthalmic, scientific community.

We also agree with the limitations of this study. Therefore we describe them in the part of the discussion, marked as “limitations of the study”.

Reviewer 2 Report

This study examined ocular findings in different types of ichthyoses. Five patients were included, and various imaging techniques were used. The results showed dry eye disease, irregular corneas, and corneal abnormalities. Multimodal imaging helped detect preclinical abnormalities and guide early examination for related disorders in ichthyosis patients.

Introduction

The introduction provides a detailed description of various forms of ichthyosis but fails to provide a broader context or rationale for studying ocular involvement in ichthyosis.

Although the introduction mentions limited reports on ocular associations with ichthyosis, it lacks a comprehensive review of existing literature.

The introduction does not clearly state the research gap or the need for the current study.

The introduction does not explicitly state the specific objectives or research questions that the study aims to address.

The introduction briefly mentions the use of multimodal imaging techniques but does not provide a strong rationale for their selection or explain how they contribute to a better understanding of ocular involvement in ichthyosis.

The description of anterior eye segment optical coherence tomography (AS OCT) and in vivo confocal microscopy (IVCM) is limited and lacks crucial details, such as their specific applications and advantages in studying ocular involvement in ichthyosis.

Method

The method section does not provide a clear rationale or power analysis to justify the sample size of the study.

The criteria for participant selection are briefly mentioned, but there is no description of how participants were recruited or whether there was any randomization process.

The exclusion criteria listed are insufficiently described. It would be helpful to provide specific details on the ophthalmic or systemic diseases that were considered exclusion criteria and how they were assessed.

The method section mentions the use of tear break-up time (TBUT) and fluorescein staining for ocular surface assessment but does not provide information on the standardized protocol used or the specific criteria for classifying abnormalities.

The method section briefly mentions the SS-OCT parameters evaluated but does not provide sufficient details on their specific definitions or clinical significance.

The method section lacks crucial details regarding the IVCM procedure, such as the specific regions of the cornea imaged, the duration of imaging, and the number of images obtained per participant.

method section does not mention whether multiple observers or the same observer performed the assessments and whether any reliability assessments were conducted.

While the study states that it was conducted in accordance with the Declaration of Helsinki and approved by the Bioethical Commission, there is no mention of how ethical considerations, such as participant privacy, confidentiality, and potential risks, were addressed.

Author Response

Thank you very much for your valuable comments and clues.

  1. Introduction

We added the following explanation and rationale for undertaking this study. We rewrote some sentences to underline the rationale for conducting a multimodal assessment of ichthyosis.

Line 34-39: Ocular involvement may be regarded as a significant feature of ichthyoses, however, the manifestation and severity of ocular signs depend on and are related to the ichthyosis clinical form. The prevalence of specific forms and subsequently ocular surface involvement vary significantly among ichthyoses. According to the prevalence the general characteristics of ichthyoses is presented.

Line 91-95: The knowledge regarding ocular complications is also limited due to the extreme rarity of particular ichthyosis variants as well as the limited lifespan of patients affected with syndromic forms. The slit-lamp examination does not provide a profound examination of the ocular surface and the corneal layer, therefore the current knowledge on the ocular complications may be limited. 

Line 100-105: The experience gained based on corneal dystrophy analysis may be complemented in ichthyoses to provide comprehensive assessment in terms of topography, morphology, and cellular imaging. To date, multimodal imaging was only described for specific ichthyosis related syndromes, such as PPCD [8-11]. Two imaging techniques play a crucial role in ocular surface assessment.  

Line 124-127: By combining these imaging methods, we aimed to characterize the appearance of the ocular surface, determine the depth of changes, and provide a profound understanding of the features associated with different variants of ichthyosis.

Commenting on the current utility of SS-OCT an IVCM in ichthyoses, we added the following sentences:

Line 112-113:  To date, no published data are available regarding the SS-OCT findings in patients with ichthyosis.

Line 118-119: To date the utility of IVCM in ichthyoses was limited to PPCD in RXLI [8-11].

  1. Methodology

Sample size

We understand, that it is a significant limitation of the study. The sample size was limited due to the extreme rarity of the disease. There was no randomization process.  Additionally, this was not a screening study. The study group consisted of patients referred to an ophthalmologist.

We addressed those issues in the part “limitations of the study” Lines 505-518.

We also added the following sentence:

Line 141: The study group consisted of patients referred from a dermatological clinic or a general practitioner.

The exclusion criteria

We added specified exclusion criteria.

Lines 144-150. Examples of such ocular diseases include corneal dystrophies (except PPCD) and degenerations, uveitis, allergic conjunctivitis, active infectious keratitis, scleritis, ocular tumors; examples of surgeries: keratoplasty, cataract surgery, pars plana vitrectomy; examples of systemic diseases: Graves orbitopathy, cystinosis, neurofibromatosis.

Ocular surface assessment

The method section including the use of tear break-up time (TBUT) and fluorescein staining for ocular surface assessment was rewritten to clarify the details of the methodology (Line 171-176).

SS OCT

We referred the reader to the previous publication focused on SS-OCT parameters assessment in normal eyes. The publication contains point by point description of each parameter. (Line 211-212).

Additional required information regarding the quality check was also provided (Line 185-188)

IVCM

Additional information was provided (Line 219-220).

Ethical issues

The Bioethical Commission of Silesian Medical University in Katowice, Poland (KNE/0022/KB1/43/I/14) approved the study protocol.

Before entering the study, patients were provided with the written “Informed Consent” and had an interview with a qualified member of the research team, during which they had the opportunity to ask any questions. After taking as much time as required, the patients signed the informed consent to participate in the study. Patients were able to withdraw from the study at any time. All study participants were covered by the hospital's third-party liability insurance. All examinations were conducted in accordance with the principles of good clinical practice applicable at the Chair and Clinical Department of Ophthalmology, Faculty of Medical Sciences in Zabrze, Medical University of Silesia, Katowice, Poland. The appropriate anonymization of personal information in the files was based on the assigned patient ID number replacing his/her name. The data was be saved in the text (doc.) or graphic format (tiff or jpg files). The acquired data and its processing was performed digitally using Microsoft Office software and its capabilities. Data was processed by qualified research personnel using validated methods. Manually entered data was checked by a second person to avoid mistypes. The data sets were organized in folders identified by the unique subject’s ID number.

Reviewer 3 Report

Dear Authors, I read with interest your article 

It explores all the ophthalmological features of Ichthyosis in a very deep and organized manner. 

Ichthyosis are a family of inherited disorders not so commonly seen in clinical practice, but with various degrees of ocular involvement; it's useful to give attention to such pathologies to help clinicians to  quickly and early recognize the affected patients. 

The paper is very well written and , although the poor number of patients enrolled in the study, the description of methods and analysis is very accurate and allows the reader to clearly understand. 

Only few questions: 

- what was the observation period of the study?

- did any of the patients involved progress to severe ocular damage during the study ?

Author Response

Thank you for your valuable comments.

We agree that the most significant limitation of our study is the low number of patients in the study group. It is mainly to the extreme rarity of the disease. We included this matter in the part of the discussion concerning the limitations of the study.

“- what was the observation period of the study?

- did any of the patients involved progress to severe ocular damage during the study ?”

The observation time was relatively short and lasted for 12 months. During this period, no adverse events, such as corneal perforation, or infectious keratitis were recorded. The automated measurements were performed once after the qualifications of patients to the study group and after signing the informed consent. We fully agree, that, providing a serial assessment would be valuable, especially focused on a long-term observation period. The patients remain under observation at our clinic and would be very valuable to reexamine the study group after a significant period of time. Taking into consideration the nature of the advanced corneal changes, such as scarring, and thinning,  a subsequent perforation is one of the major possibilities, that could occur.

We also included this as one of the limitations of the study. No reliable statistical analyses were possible for this small and diverse study group.  (Line 512).

Reviewer 4 Report

Dear Authors,

Well presented and interesting clinical paper.

New imaging modalities were used to examine eyes of patients with ichthyosis,

Results could be important for further clinical practice, however examined group is small.

Minor issues:

line 14: i would not describe difference of 5 or 10 µm between CAT and CTT as significant

line 181: no scale for BCVA is given (decimal?), abbrevation is not explained

Best regards

Author Response

Thank you very much for your comments.

Line 14.

We fully agree, that the difference of 5 um is not significant. We provided a range, but on the other hand, aimed to underline, that the difference exceeded to 375 um.

We corrected this sentence.

BCVA.

Thank you for the comment. We explained the abbreviation in the text. Yes, it is decimal. BCVA is measured from five meters in Poland and the decimal scale of the Snellen chart is the most popular one. (Line 223). We also added the appropriate explanation in Table 2.

Round 2

Reviewer 2 Report

I appreciate the efforts you have taken in revising the manuscript based on the feedback provided. However, upon careful review of the updated submission, I find that the changes made in this second round of revision are still not sufficient to address the primary concerns raised. It is essential for the integrity and quality of the work that these concerns are fully addressed to meet the journal's standards. 

Author Response

Dear Reviewer,

Thank you for your comments. We made an effort to carefully revise and make additional changes to the manuscript.

Here, we present listed new changes (round 2) implemented to the manuscript:

  1. Lines 104-107. We provided an additional explanation of the rationale behind this report.

“Describing the morphological features of ichthyoses including the character of abnormalities, anterior and posterior keratometry values, and morphology of all corneal layers, using non-invasive imaging methods would bring new valuable data in the context of the impact of ichthyoses on the corneal structure”.

  1. Lines 139-144. We provided additional data regarding the ethical concerns raised by the reviewer.

“Patients were able to withdraw from the study at any time. All study participants were covered by the hospital's third-party liability insurance. All examinations were conducted in accordance with the principles of good clinical practice applicable at the Chair and Clinical Department of Ophthalmology, Faculty of Medical Sciences in Zabrze, Medical University of Silesia, Katowice, Poland. The appropriate anonymization of personal information in the files was based on the assigned patient ID number”. 

  1. Regarding “The method section briefly mentions the SS-OCT parameters evaluated but does not provide sufficient details on their specific definitions or clinical significance”.

We provided an additional Table S1 with all OCT parameters analyzed in the study clearly explained.

Parameter

Development of an English Abbreviation

Description of the Parameter

Unit

Corneal Parameters [k, keratometric; r, real; p, posterior]

Ks

Keratometry steep

Steep meridian value of keratometry of the curvature of the cornea

[D]—Diopter

Kf

Keratometry flat

Flat meridian value of keratometry of the curvature of the cornea

[D]—Diopter

CYL

Cylinder

Astigmatism power

[D]—Diopter

AvgK

Average Keratometry

Mean value of steep meridian Ks and flat meridian Kf

[D]—Diopter

Ecc

Eccentricity corneal curve

Ecc corresponds to the numeric value of the eccentricity at a 9 mm diameter area. Eccentricity describes the rate of corneal flattening from the central cornea to the periphery

Numerical range 0-1

AA [%]

Area Analyzed

AA reflects the percentage of performed automatic analysis at a 10 mm diameter area [assuming the whole circumference as 100%]

[%] percentage

ACCP

Average Central Corneal Power

The mean value of axial power within a 3 mm diameter

[D]—Diopter

CAT

Central Apical Thickness

The corneal thickness on the measurement axis

[µm]—micrometers

CTT

Corneal Thinnest Thickness

Corneal thinnest thickness

[µm]—micrometers

CAT-CTT

Central Apical Thickness minus Corneal Thinnest Thickness

The difference between central apical thickness and corneal thinnest thickness

[µm]—micrometers

ACD [mm]

Anterior Chamber Depth

Anterior chamber depth from the posterior corneal surface to the anterior crystalline lens surface

[mm] millimeters

ESI [%]

Ectasia Screening Index

ESI (%) is assessed based on the following data: the thinnest corneal thickness (μm) and its location related to the corneal apex (mm) (X, Y coordinates); FI of the keratometric and posterior topography data; the location and the lowest value of the instantaneous posterior power within a 6 mm diameter.

[%] percentage

Fourier indices; area of 3 mm and 6 mm; k, keratometric; a, anterior; p, posterior

Spherical

Fourier Index Spherical

The spherical refractive power component of the cornea obtained by Fourier analysis of the topographic data of the cornea with a diameter of 3 and 6 mm. Corresponds to the zero-order component in Fourier analysis

[D]—Diopter

Reg. Astigmatism

Fourier Index Regular

The component of regular corneal refractive power astigmatism obtained by Fourier analysis of corneal topographic data with a diameter of 3 and 6 mm. Corresponds to the second-order component in the Fourier analysis

[D]—Diopter

Asymmetry

Fourier Index Asymmetry

Asymmetric refractive power component of the cornea obtained by Fourier analysis of corneal topographic data with a diameter of 3 and 6 mm. Corresponds to the first-order component in Fourier analysis

[D]—Diopter

Higher Order

Fourier Index Higher Order

Higher order irregularities—calculated by combining tertiary and higher components in Fourier analysis

[D]—Diopter

  1. Lines 227-232. We provided additional description regarding “The method section lacks crucial details regarding the IVCM procedure, such as the specific regions of the cornea imaged, the duration of imaging, and the number of images obtained per participant”.

“This exam was the last performed after ocular surface assessment and OCT, due to its invasive character. The central corneal part was assessed (3-4 mm diameter), the mean examination time was 5 minutes per eye and the mean number of acquired images was 15 scans per eye. The examination was performed by one observer (JKL) and the validation of the results was confirmed by the second observer (AN). All scans were analyzed and the representative scans presenting abnormalities were chosen”.

Regards

Anna Nowinska